# Evaluating the Robustness of Chinchilla Compute-Optimal Scaling

## Abstract

Hoffmann et al. (2022)'s Chinchilla paper introduced the principle of compute-optimal scaling, laying a foundation for future scaling of language models. In the years since, however, valid concerns about Chinchilla have been raised: wide confidence intervals, discrepancies between its three approaches, and incongruities with other scaling laws. This raises a critical question for the field: Can practitioners still rely on Chinchilla's prescriptions? Our work demonstrates the answer is yes. We begin by uncovering that the model parameters central to Chinchilla's analyses were ambiguous: three interpretations are possible, with relative differences between different interpretations of model parameters as high as 15.2%. We find that, perhaps surprisingly, which model parameters are used for the analyses do not meaningfully affect key results: the scaling law estimates and the compute-optimal tokens-to-parameter ratio. Indeed, under one interpretation, the tokens-to-parameter ratio becomes more constant with the target compute budget. We then ask how distorted the Chinchilla model parameters *could* have been without meaningfully affecting the key results. By deliberately perturbing model parameters in four structured ways, we find that key Chinchilla results are most sensitive to additive or systematic errors, which can alter the otherwise flat trend of the optimal tokens-to-parameter ratio, but overall, Chinchilla's key results withstand sizable perturbations. Altogether, our findings offer the field renewed confidence in Chinchilla as a durable guide for scaling language models.

## 1 Introduction

The study of neural scaling laws, which predictably map training resources to model performance, is a cornerstone of modern language modeling research and engineering. Hestness et al. (2017) and soon after Kaplan et al. (2020) laid the foundation by demonstrating that pretraining losses scale as power laws with the number of data points, model parameters and pretraining compute. This was followed by significant additional work in the ensuing years (Sec. 4 Related Work). Such discoveries led to the modern paradigm of training extremely large language models (OpenAI et al., 2024; DeepMind et al., 2025; DeepSeek-AI et al., 2025; Qwen et al., 2025; Kimi et al., 2025; Zhipu-AI et al., 2025).

The field's understanding of scaling models was later altered by the seminal work of Hoffmann et al. (2022), which introduced the concept of compute-optimal scaling. By training over 400 models ranging from 44M to 16B parameters on 5B to 500B tokens, Hoffmann et al. (2022) discovered that producing the best performing model with respect to a fixed pretraining compute budget ("compute-optimal") was achieved by linearly scaling model parameters and pretraining data together. Chinchilla established the influential "20-to-1" heuristic: that the compute-optimal amount of training data is approximately 20 tokens per model parameter (Appendix C). Their 70B Chinchilla outperformed larger models (Rae et al., 2022), cementing the methodology as a guiding principle for the field.

In the years since, several contributions have closely scrutinized Chinchilla, raising a number of concerns: Zhang (2023) called attention to Chinchilla's wide confidence intervals and questioned whether such uncertain estimates can provide practical guidance. Besiroglu et al. (2024) investigated why some of Chinchilla's approaches yielded inconsistent results. Lastly, Porian et al. (2024) and Pearce & Song (2024) examined why Chinchilla makes different predictions than Kaplan et al. (2020)'s earlier scaling work. While these works are clear contributions to the science of scaling, the field has been left uncertain: can practitioners still confidently rely on Chinchilla's prescriptions?

In this work, we aim to answer this question. As a warm up, we uncover that the model parameters central to the Chinchilla analyses were ambiguous, with three different possible interpretations as to which model parameters were used: (1) the model parameters reported in Hoffmann et al. (2022)'s Table A9, (2) the model parameters calculated from the reported model architectural hyperparameters (layers, dimensions, number of heads, etc.) using a "standard" formula, and (3) the model parameters calculated from a "best-fit" formula. Although the relative error among these three sets of model parameters rises as high as 15.2%, we show that key Chinchilla results – the estimated scaling law parameters and the compute-optimal tokens-per-parameter ratio – do not meaningfully change. In fact, the only potential consequence is that the compute-optimal tokens-per-parameter ratio becomes *more* constant with respect to the target compute budget, strengthening Chinchilla's finding.

To more generally assess the robustness of compute-optimal scaling, we then study how distorted the model parameters *could* have been without changing Chinchilla's key results. We perform a sensitivity analysis by perturbing model parameters in four structured ways. Our analyses reveals that the robustness depends on the nature of the perturbations: while multiplicative perturbations and random noise have limited effects, additive constants or systematic biases can qualitatively change the compute-optimal scaling strategy by altering the trend of the optimal tokens-to-parameter ratio. However, overall, all four sensitivity analyses demonstrate that Chinchilla's key results withstand sizable perturbations. Our results reveal a clear picture: Chinchilla's compute-optimal prescription remains robust, further justifying its widespread use as a practical scaling blueprint for practitioners.

## 2 KEY CHINCHILLA RESULTS ARE ROBUST TO THREE INTERPRETATIONS OF CHINCHILLA MODEL PARAMETERS

One of the fundamental inputs to the Chinchilla analyses are the number of parameters per model. However, an ambiguity exists as to which exact model parameters were used, with three different possible interpretations differing by as much as 15.2%. We uncovered this by closely examining Chinchilla's Table A9, which reports the number of model parameters for each model alongside key architectural hyperparameters, e.g., d_model, ffw_size, kv_size, n_heads and n_layers. We call the model parameters reported in Chinchilla's Table A9 the **reported model parameters**. We include a brief snippet in our main text (Table 1) and the full table in our Appendix B.

However, a second interpretation of the model parameters arises from the provided architecture hyperparameters; assuming the embedding and unembedding weights are tied (Press & Wolf, 2017) and no gating is present, a standard formula for the number of model parameters is:

$$
\begin{aligned}
\text{Standard Formula Model Params} &\approx \text{Embedding Params} + \text{Attn Params} + \text{FFN Params} \\
\text{Embedding Params} &= \text{Vocab Size} \cdot \text{d\_model} \\
\text{Attn Params} &= \text{n\_layers} \cdot (4 \cdot \text{d\_model} \cdot \text{kv\_size} \cdot \text{n\_heads}) \\
\text{FFN Params} &= \text{n\_layers} \cdot (2 \cdot \text{d\_model} \cdot \text{ffw\_size})
\end{aligned}
\tag{1}
$$

Comparing the **standard formula model parameters** with the reported model parameters reveals a mismatch for *every* model, with an average relative error of 7.4% but reaching as high as 15.2% and no less than 3.6% (Fig. 1, left). We calculate relative error as:

$$
\text{Relative Error (\%)} = 100 \cdot \frac{\text{Reported Model Params} - \text{Standard Formula Model Params}}{\text{Reported Model Params}}.
\tag{2}
$$

In an attempt to reconcile the two interpretations of model parameters, we determined a third interpretation based on a "best fit" formula that nearly matches the reported model parameters:

$$
\begin{aligned}
\text{Best Fit Formula Model Params} &\approx \text{Embedding Params} + \text{Attn Params} + \text{FFN Params} \\
\text{Embedding Params} &= \text{Vocab Size} \cdot \text{d\_model} \\
\text{Attn Params} &= \text{n\_layers} \cdot (\mathbf{5} \cdot \text{d\_model} \cdot \text{kv\_size} \cdot \text{n\_heads}) \\
\text{FFN Params} &= \text{n\_layers} \cdot (2 \cdot \text{d\_model} \cdot \text{ffw\_size})
\end{aligned}
\tag{3}
$$

Switching from the standard formula model parameters to the **best fit model parameters** reduced the number of discrepancies with the reported model parameters from $50/50$ models to $6/50$ models, and reduced the largest relative error from 15.2% to 8.7% (Fig. 1, right).

| Table A9 from Hoffmann et al. (2022) | | | | | | Our Contribution | | |
|---|---|---|---|---|---|---|---|---|
| d_model | ffw_size | kv_size | n_heads | n_layers | n_vocab | Chinchilla's Reported Model Parameters (M) | Best Fit Formula's Model Parameters (M) | Standard Formula's Model Parameters (M) |
| 512 | 2048 | 64 | 8 | 8 | 32168 | 44 | 44 | 42 |
| 576 | 2304 | 64 | 9 | 9 | 32168 | 57 | 57 | 54 |
| 640 | 2560 | 64 | 10 | 10 | 32168 | 74 | 74 | 70 |
| ⋮ | ⋮ | ⋮ | ⋮ | ⋮ | ⋮ | ⋮ | ⋮ | ⋮ |
| 4864 | 19456 | 128 | 36 | 47 | 32168 | 13775 | 14319 | 13266 |
| 4992 | 19968 | 128 | 32 | 49 | 32168 | 14940 | 14939 | 13937 |
| 5120 | 20480 | 128 | 40 | 47 | 32168 | 16183 | 16182 | 14950 |

Table 1: **Three Interpretations of Chinchilla's Model Parameters.** Hoffmann et al. (2022)'s Table A9 provides the architectural hyperparameters of all models used in the Chinchilla analyses, along with the reported model parameters (specified in millions). However, two alternative interpretations of model parameters are possible: model parameters calculated from architectural hyperparameters using a "standard" formula (Eqn. 1) and model parameters calculated from architectural hyperparameters using a "best fit" formula (Eqn. 3). For the complete table, see Appendix B.

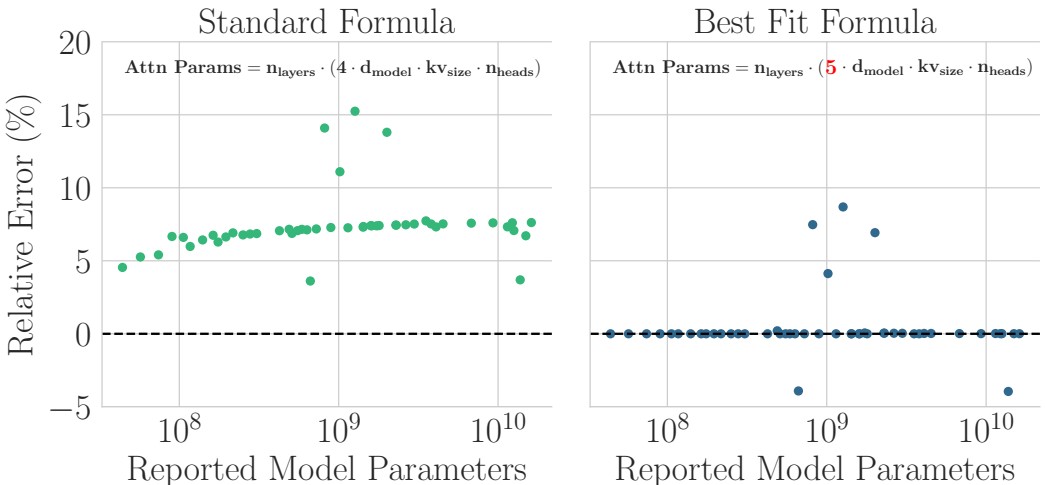

Figure 1: **Disagreement Between Three Interpretations of Chinchilla's Model Parameters.** Each point is one of the 50 models in Hoffmann et al. (2022)'s Table A9. **Left:** Calculating model parameters from the provided architectural hyperparameters using a *standard formula* (Eqn. 1; attention parameters = n_layers $\cdot$ 4 $\cdot$ d_model $\cdot$ kv_size $\cdot$ n_heads) disagrees with the reported model parameters for $50/50$ models, with relative errors averaging $7.388\%$ and rising as high as $15.2\%$. **Right:** Calculating model parameters using a *best fit formula* (Eqn. 3; replace 4 with 5) matches $44/50$ of the reported model parameters, and reduces the largest relative error to $8.7\%$.

Having identified these three different notions of model parameters, we tested how Chinchilla's key results might change depending on which model parameters are used for fitting. We focus on two key results in particular: First, Chinchilla fit a neural scaling law to the pretraining loss $L$ as a function of model parameters $N$ and pretraining data $D$:

$$L(N, D) = E + \frac{A}{N^\alpha} + \frac{B}{D^\beta}, \tag{4}$$

where $E$ is the irreducible error, $A$ is the parameter prefactor, $\alpha$ is the parameter exponent, $B$ is the data prefactor and $\beta$ is the data exponent. Second, Chinchilla derived from the estimated scaling law parameters a "20-to-1" heuristic for the compute-optimal ratio of tokens-per-parameters:

$$\text{Compute-Optimal Tokens-per-Parameter Ratio} \approx 20. \tag{5}$$

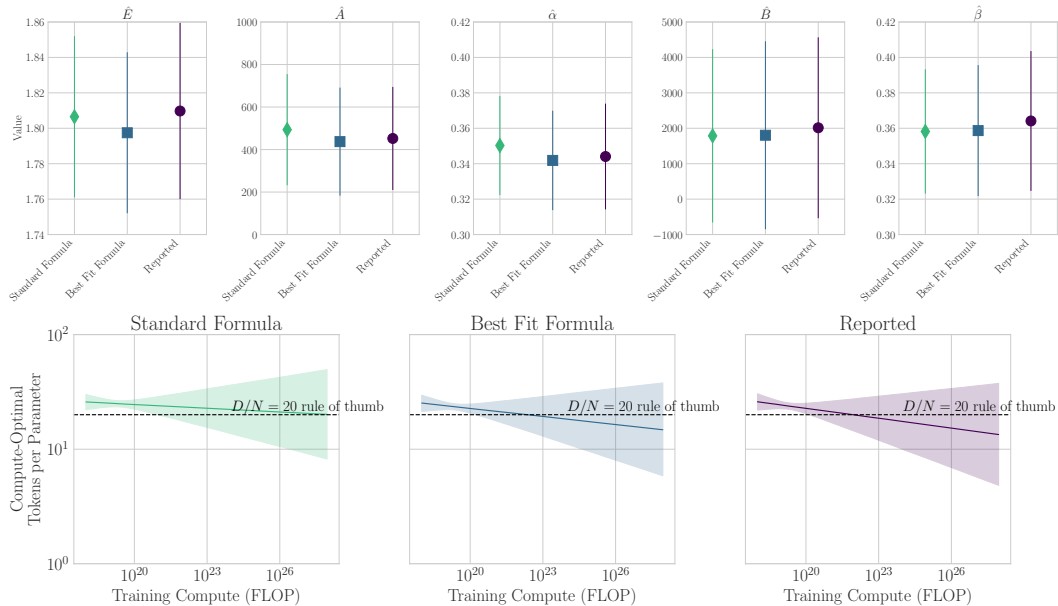

Figure 2: **Key Chinchilla Results Are Robust to All Three Interpretations of Model Parameters.** Hoffmann et al. (2022) fit a neural scaling law $L(N, D) = E + A \cdot N^{-\alpha} + B \cdot D^{-\beta}$, where $N$ is the number of model parameters and $D$ is the number of data (Eqn. 4). **Top:** The fit parameters $(\hat{E}, \hat{A}, \hat{\alpha}, \hat{B}, \hat{\beta})$ do not meaningfully change, regardless of which model parameters are used for fitting. Error bars are standard errors from 4000 bootstrapped samples. **Bottom:** The compute-optimal tokens-per-parameter ratio remains constant at $\approx 20$, regardless of which notion of model parameters are used in the fitting process. The slope is flattest with the standard formula model parameters ($-0.572$ per decade; best fit: $-1.049$; reported: $-1.248$). Error bars are 80% confidence intervals. Fitting and visualization were conducted using Besiroglu et al. (2024)'s code.

We tested how these two Chinchilla headline results change depending on which of the three interpretations is used in the fits: (1) Reported Model Parameters, (2) Standard Formula Model Parameters, or (3) Best Fit Formula Model Parameters. To conduct the Chinchilla fits and visualize the results, we used Besiroglu et al. (2024)'s Chinchilla fitting code that was previously used to align Chinchilla's three approaches.

Perhaps surprisingly, we found that none of the five fit parameters $\hat{E}, \hat{A}, \hat{\alpha}, \hat{B}, \hat{\beta}$ differed significantly depending on which of our three notions of model parameters were used in fitting (Fig. 2, top). We similarly found the compute-optimal tokens-per-parameter ratio remains constant around 20 tokens per parameter (Fig. 2, bottom). Arguably, the standard formula model parameters yield a *flatter* trend with increasing training compute: the slope for the standard formula model parameters is $-0.572$ for each 10x increase in compute, which decreased to $-1.049$ for the best fit formula model parameters and decreased further to $-1.248$ for the reported model parameters. However, uncertainty makes drawing strong conclusions difficult. These results demonstrate that **key Chinchilla results are robust to whichever of our three notions of model parameters is used in the fitting process**.

## 3 ROBUSTNESS OF CHINCHILLA HEADLINE RESULTS DEPENDS ON TYPE OF PERTURBATION TO MODEL PARAMETERS

Given that the key Chinchilla results did not meaningfully change even when model parameters differed by as much as 15.2%, we next asked:

> *How distorted could the model parameters have been without meaningfully affecting Chinchilla's headline results?*

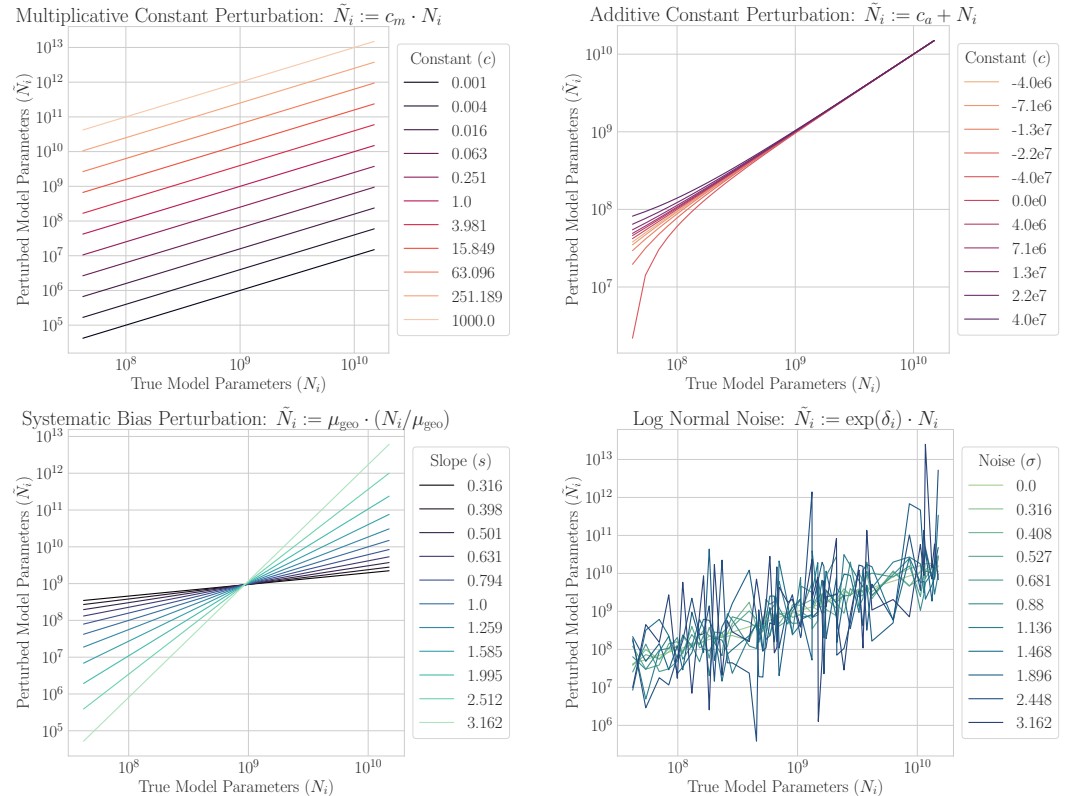

Figure 3: **Evaluating the Robustness of Chinchilla via Four Model Parameter Perturbations.**
We study how robust key Chinchilla results are to structured perturbations of models' parameters.
**Top Left:** In the first perturbation, motivated by Sec. 2, we perturb model parameters with a multiplicative constant $c_m$. **Top Right:** In the second perturbation, we perturb model parameters with an additive constant $c_a$, that could perhaps arise due to embedding parameters being included/excluded.
**Bottom Left:** In the third perturbation, we perturb model parameters with a systematic bias: either smaller models' parameters are larger and larger models' parameters are smaller, or smaller models' parameters are smaller and larger models' parameters are larger; the systematic bias is controlled by slope $s$. **Bottom Right:** In the fourth perturbation, we assume the relationship of the loss with model parameters is perhaps noisy, e.g., (Frankle & Carbin, 2019), with noise strength parameterized by $\sigma$.

To answer this question, we intentionally perturbed the standard formula model parameters in four structured ways: multiplicative constant, additive constant, systematic bias and log normal noise. We then reran the Chinchilla fitting processes using the perturbed model parameters to see what effect each type of perturbation has on the estimated scaling law parameters and compute-optimal tokens-per-parameter. We offer visual intuition for each of the four types of perturbations in Fig. 3, and mathematically describe each type of perturbation along with its consequences below. For our theoretical analyses, see Appendix C.

## 3.1 MULTIPLICATIVE CONSTANT PERTURBATION INCREASES $\hat{A}$ EXPONENTIALLY

Motivated by Sec. 2, for our first perturbation, we assume model parameters are systematically under/overestimated by approximately the same percentage. To model this, we multiplied all true model parameters $\{N_i\}_i$ by constant multiplier $c_m$ to produce perturbed model parameters $\{\tilde{N}_i\}_i$:

$$\tilde{N}_i \stackrel{\text{def}}{=} c_m \cdot N_i. \tag{6}$$

We swept $c_m$ in logspace(-3, 3, num=11). For visual intuition, see Fig. 3, top left.

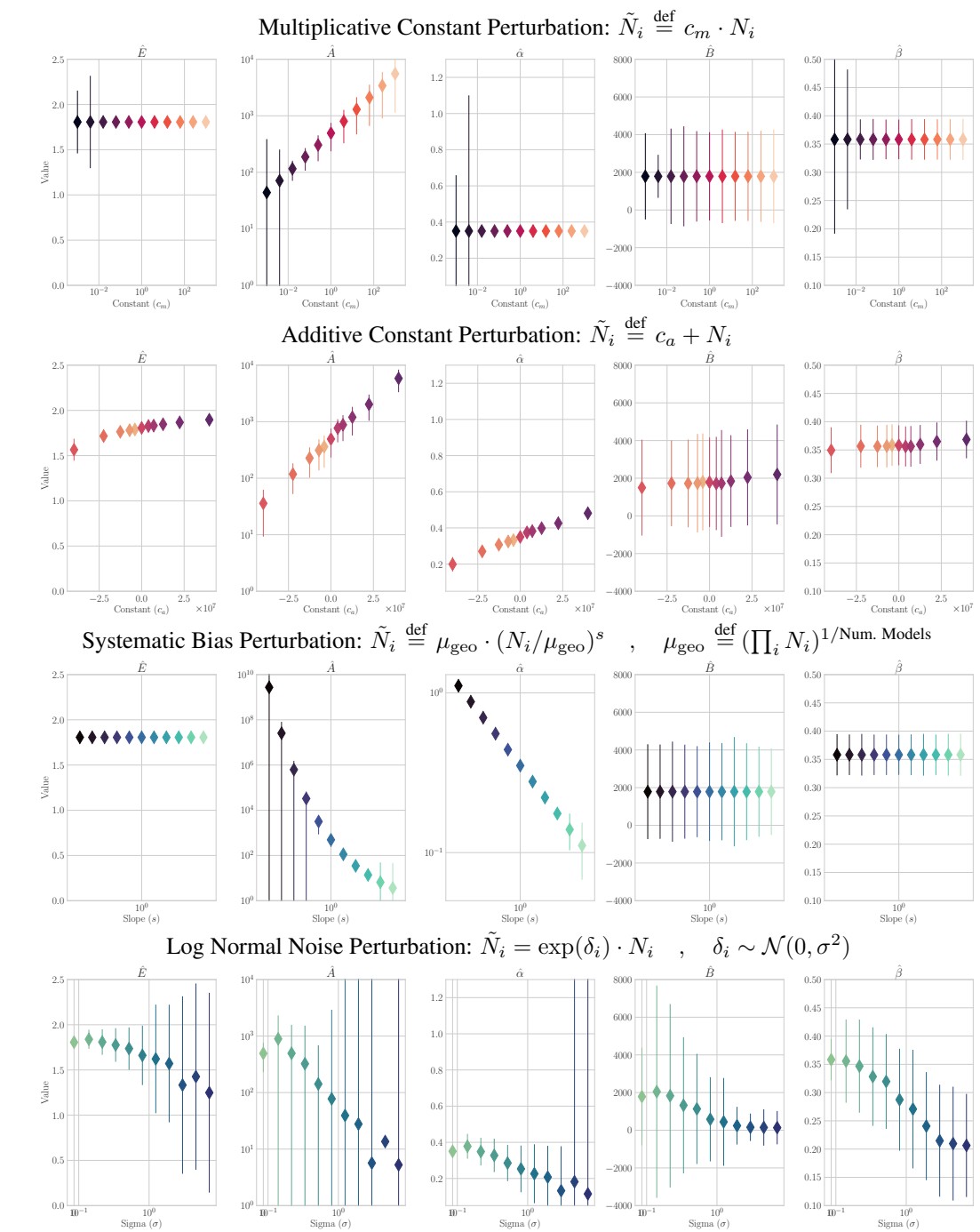

Figure 4: **Robustness of Fit Neural Scaling Law Parameters Under Four Types of Model Parameter Perturbations.** Each row visualizes the effect of a different perturbation on the five fit parameters of the Chinchilla scaling law $(L(N, D) = E + A \cdot N^{-\alpha} + B \cdot D^{-\beta})$. **Row 1:** A multiplicative constant perturbation $(c)$ increases the model parameter prefactor $(\hat{A})$ exponentially, while other fit parameters remain stable. **Row 2:** An additive constant perturbation $(c)$ linearly increases the model parameter exponent $(\hat{\alpha})$ and exponentially increases its prefactor $(\hat{A})$, with only a gentle rise in the irreducible loss $(\hat{E})$. **Row 3:** A systematic bias perturbation $(s)$ causes the model parameter exponent $(\hat{\alpha})$ to decay as a power law $(s^{-1})$ and the prefactor $(\hat{A})$ to decline sub-polynomially. **Row 4:** Adding log-normal noise $(\sigma)$ primarily increases the uncertainty of all fit parameters, while weakly decreasing the model parameter exponent $(\hat{\alpha})$ logarithmically and the prefactor $(\hat{A})$ polynomially. Error bars are standard errors obtained by $4000$ bootstrapped samples.

As we derive in Appendix C.2.1 and show empirically in Fig. 4's first row, the fitting process compensates for this multiplicative error primarily by adjusting the model size prefactor, $\tilde{\tilde{A}}$, to approximately $\hat{A}c_m^\alpha$, while the scaling exponent, $\hat{\alpha}$, remains largely unchanged, i.e., $\tilde{\tilde{\alpha}} \approx \hat{\alpha}$. This makes sense for moderate $c_m$: if $\hat{A}$ is the best fit for the true model parameters, then replacing the true parameters with the perturbed parameters $N_i \to \tilde{N}_i$ and rescaling $\hat{A} \to \tilde{\tilde{A}} = (c_m)^\alpha \hat{A}$ produces approximately the same fit. As a consequence, Fig. 5 Top Left shows the compute-optimal tokens per parameter remains constant with pretraining compute, but the precise constant grows as a power of $c_m$ if true model parameters are underestimated ($c_m < 1$) and shrinks if true model parameters are overestimated ($c_m > 1$). The only exceptions are for the two smallest multiplicative constants (0.001 and 0.004), where uncertainty in $\hat{A}$ and $\hat{\alpha}$ produced `NaNs`.

### 3.2 Additive Constant Perturbation Increases $\hat{\alpha}$ Linearly and $\hat{A}$ Exponentially

In our second perturbation, we assume model parameters have an additive term. For example, embedding parameters may be included or excluded, a key detail in previous scaling law studies (Kaplan et al., 2020; Hoffmann et al., 2022) that is partially responsible for discrepancies between estimated scaling laws' parameters (Pearce & Song, 2024; Porian et al., 2024)). To model this, we added constant $c_a$ to all model parameters:

$$\tilde{N}_i \stackrel{\text{def}}{=} c_a + N_i. \tag{7}$$

We swept $c_a$ in -logspace(6.6, 7.6, num=5) $\cup \{0\} \cup$ logspace(6.6, 7.6, num=5). For visual intuition, see Fig. 3 Top Right. For additional context, the smallest Chinchilla model has $42 \times 10^6$ parameters.

Fig. 4's second row shows the effects: *(i)* the irreducible loss $\hat{E}$ rises only gently from 1.565 to 1.897 ($\approx 21\%$). *(ii)* the model parameter prefactor $\hat{A}$ grows exponentially in $c_a$, increasing by $\sim 2.5x$ from the most negative to the most positive constant *(iii)* the model parameter exponent $\hat{\alpha}$ increases linearly with $c_a$ from 0.199 to 0.481 and *(iv)* both the data prefactor $\hat{B}$ and data exponent $\hat{\beta}$ fluctuate only within their bootstrap error bars and show no systematic trend. As a consequence, Fig. 5 Top Right shows the compute-optimal tokens per parameter becomes less constant with the training compute: a larger positive $c_a$ means larger target training horizons require more tokens per parameter, whereas a larger negative $c_a$ means larger target training horizons require fewer tokens per parameter.

In Appendix C.2.2, we analytically explain these trends: the most critical parameter in a power law is its exponent, which corresponds to its slope in log-log space. However, for the perturbed function, the slope is now no longer constant and depends on $N$ as $N/(N + c_a)$. Thus, the fitting procedure must select a single exponent that best represents the varying slope over the range of data. When $c_a > 0$, the factor $N/(N + c_a) < 1$, and the fitting process must select an exponent $\tilde{\tilde{\alpha}} > \hat{\alpha}$; and when $c_a < 0$, the factor $N/(N + c_a) > 1$, and to compensate, the fitting process must select an exponent $\tilde{\tilde{\alpha}} < \hat{\alpha}$.

For comparison, Porian et al. (2024) found that including the model's head parameters increased the fit model parameter scaling exponent $\hat{\alpha}$ by $0.080\,(0.072 \to 0.152)$, and Pearce & Song (2024) found that including embedding parameters increased $\hat{\alpha}$ by $0.231\,(0.135 \to 0.366)$. Although assuming an additive constant is a simplification of both analyses, all three results are quantitatively similar.

### 3.3 Systematic Bias Perturbation Decreases $\hat{\alpha}$ Polynomially and $\hat{A}$ Sub-Polynomially

In our third perturbation, we assume the presence of a systematic bias in reported models' parameters: either the smaller models' parameters are truly larger and the larger models' parameters truly smaller, or vice versa. To model this, we define the perturbed parameters as

$$\tilde{N}_i \stackrel{\text{def}}{=} \mu_{\text{geo}} \cdot (N_i / \mu_{\text{geo}})^s, \tag{8}$$

where $\mu_{\text{geo}} \stackrel{\text{def}}{=} (\prod_i N_i)^{1/\text{Num. Models}}$ is the geometric mean of the model parameters and $s$ is the systematic bias parameter: $s < 1$ shrinks large models and inflates small ones, whereas $s > 1$ does the reverse. We swept $s$ in logspace(-0.5, 0.5, 11). For visual intuition, see Fig. 3 Bottom Left.

The third row of Fig. 4 illustrates three main effects: *(i)* The model parameter exponent $\hat{\alpha}$ decays according to the power-law relationship $\hat{\alpha} = 10^{-0.46} \cdot s^{-1}$, which is a nearly perfect fit to the data

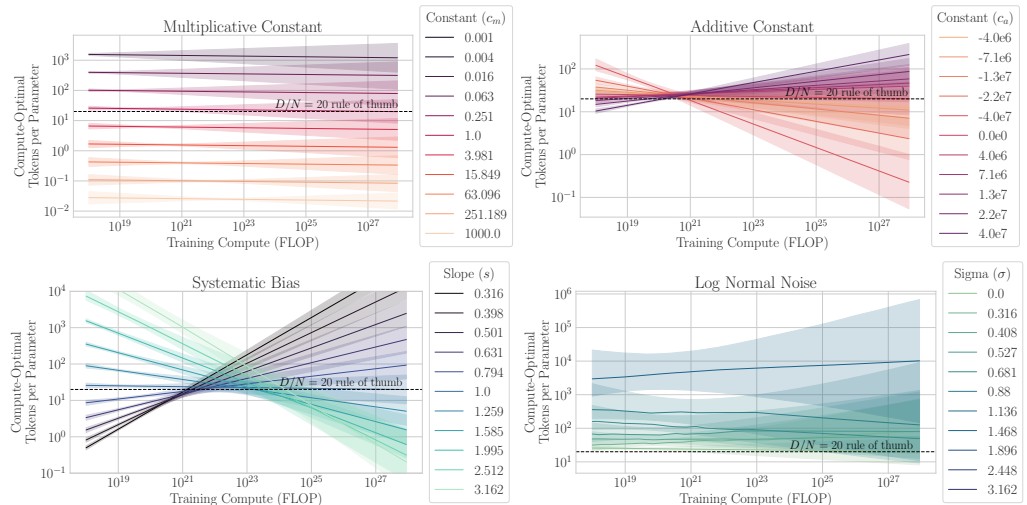

Figure 5: **Robustness of Compute-Optimal Tokens-Per-Parameter Under Four Types of Model Parameter Perturbations.** Shaded regions represent 80% confidence intervals. **Top Left:** A multiplicative constant perturbation by $c$ shifts the compute-optimal ratio by $c^\alpha$ but keeps the trend flat with respect to training compute. **Top Right:** An additive constant perturbation by $c$ makes the compute-optimal ratio less constant across the target training compute horizon. A positive slope means more tokens are needed per parameter for larger compute budgets, while a negative slope means fewer are required. **Bottom Left:** A systematic bias also makes the ratio less constant. A larger bias ($s > 1$) leads to fewer optimal tokens per parameter for larger models, whereas a smaller bias ($s < 1$) requires more. **Bottom Right:** Adding log-normal noise to model parameters increases the uncertainty and the overall magnitude of the compute-optimal tokens per parameter ratio.

($R^2 > 0.999$, $p \approx 5.9 \times 10^{-90}$). *(ii)* The model parameter prefactor $\hat{A}$ declines sub-polynomially. *(iii)* The irreducible loss, $\hat{E}$, and the data parameters, $\hat{B}$ and $\hat{\beta}$, show no systematic trend, with fluctuations remaining within their bootstrap error bars. Similar to the Additive Constant perturbation, Fig. 5 Bottom Left shows that the two trends of $\hat{A}$ and $\hat{\alpha}$ together make the compute-optimal tokens per parameter ratio less constant with the target training horizon: a larger systematic bias $s$ means larger target training horizons require fewer tokens per parameter, whereas a smaller systematic bias $s$ means larger target training horizons require more tokens per parameter.

In Appendix C.2.3, we mathematically derive that under the systematic bias, the model parameter exponent is multiplied by $s^{-1}$ and the model prefactor is multiplied by $\mu_{\text{geo}}^{\alpha(1-s)/s}$, making the exponent in the compute-optimal ratio $(\alpha/s - \beta)/(\alpha/s + \beta)$. Thus, if $s < 1$, then the exponent on $C$ becomes positive and compute-optimal ratio of tokens-per-parameter increases with compute, whereas if $s > 1$, then the exponent on C becomes negative and the compute-optimal ratio of tokens-per-parameter decreases with compute.

### 3.4 LOG NORMAL NOISE PERTURBATION INCREASES UNCERTAINTY AND DECREASES $\hat{\alpha}$ LOGARITHMICALLY AND $\hat{A}$ POLYNOMIALLY

In our fourth perturbation, we assume the "value" of model parameters is noisily measured, perhaps due to model initializations. To model this, we added log-normal noise to the number of parameters. Specifically, for each model's parameter count $N_i$, we sampled a new parameter count as:

$$\tilde{N}_i \overset{\text{def}}{=} \exp(\delta_i) \cdot N_i, \qquad \delta_i \sim \mathcal{N}(0, \sigma^2). \tag{9}$$

We swept $\sigma$ from $1 \times 10^{-2}$ to $1 \times 10^2$. For visual intuition, see Fig. 3 Bottom Right.

The fourth row of Fig. 4 illustrates three main effects: *(i)* Nearly all fit parameters have significantly larger confidence intervals, especially as the noise standard deviation $\sigma$ increases; for the highest value of 3.162, $\hat{A}$ and $\hat{\alpha}$ are nearly unidentifiable. *(ii)* To the extent that trends can be identified, the irreducible error $\hat{E}$ trends down weakly, while the model parameters prefactor $\hat{A}$ falls roughly

polynomially and the model parameters exponent falls roughly logarithmically with the noise standard deviation $\sigma$. Fig. 5 Bottom Right demonstrates the consequences of the noise: fits with too high of noise create NaNs, while noise drives up the compute-optimal tokens per parameter and also increases the width of the 80% confidence intervals by $\sim 1$ order of magnitude, although the inferred values are roughly constant with target training compute.

# 4 RELATED WORK

Due to space constraints, we defer most Related Work to Appendix D and focus here on prior research most relevant to our contribution. The precise details of Hoffmann et al. (2022) have recently come under scrutiny, leading to a number of important replication and re-evaluation studies. For instance, Chinchilla used three different approaches, two of which agreed with each other, but the third did not; Besiroglu et al. (2024) conducted a detailed investigation of this third analysis and found that it could be made consistent with the first two analyses by fixing optimizer issues and not rounding reported fit parameters. In a similar vein, Porian et al. (2024) and Pearce & Song (2024) sought to resolve a discrepancy between Hoffmann et al. (2022) and Kaplan et al. (2020) on how to scale data and parameters to produce the best performing model; Porian et al. (2024) found that the discrepancy could be resolved by three differences (last layer computational cost, warmup duration, and scale-dependent optimizer tuning) while Pearce & Song (2024) found that much of the discrepancy could be attributed to Kaplan et al. (2020) counting only non-embedding parameters.

Like Besiroglu et al. (2024) and Porian et al. (2024), our work scrutinizes the seminal work of Chinchilla. However, our analyses focuses specifically on how robust the original Chinchilla methodology and results are to different perturbations. Our contribution concludes with a direct confirmation of the original findings, providing evidence that Chinchilla's compute-optimal guidance is robust.

# 5 DISCUSSION

This work began with a perhaps surprising result: three different interpretations of Chinchilla's model parameters are possible, with discrepancies as high as 15.2%, but all three support (or strengthen) key Chinchilla results. Neither the estimated scaling law parameters nor the widely adopted "20-to-1" compute-optimal tokens-to-parameter ratio changed meaningfully. Indeed, our refitting using the standard formula model parameters suggests an even more stable relationship, with the token-to-parameter ratio varying even less across different pretraining compute budgets.

To understand this robustness more deeply, we systematically investigated how various hypothetical perturbations would affect key Chinchilla results. We perturbed the parameter counts in four structured ways and re-ran the fitting analysis for each. This stress test revealed the specific ways in which different types of errors impact the scaling law parameters. A simple multiplicative error, for example, exponentially shifts the constant in the optimal tokens-per-parameter ratio, while an additive error or a systematic bias can more dramatically alter its trend with respect to the target training compute budget.

Ultimately, our findings serve as both a critical re-examination and a powerful confirmation of the original Chinchilla results. Our subsequent analyses should give practitioners even greater confidence in Chinchilla's compute-optimal prescription. Its guidance withstands not only the specific interpretation used, but also a range of other potential perturbations, reinforcing its value as a durable and practical blueprint for the field.

**Future Directions** One obvious next step is to evaluate the robustness of more recent scaling results with additional considerations such as inference constraints (Sardana et al., 2024), data constraints (Muennighoff et al., 2023) and overtraining (Gadre et al., 2024).

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

## A  LANGUAGE MODEL USAGE

Language models were used by the authors to aid or polish the writing of the paper. Authors take full responsibility for the content.

# B  CHINCHILLA'S ARCHITECTURAL HYPERPARAMETERS AND MODEL PARAMETERS

Below, we include Hoffmann et al. (2022)'s Table A9 listing model architectural hyperparameters alongside the reported model parameters. We augment the table with the standard formula model parameters (Eqn. 1) and the best fit formula model parameters (Eqn. 3).

| Table A9 from Hoffmann et al. (2022) | | | | | | | Our Contribution | |
|---|---|---|---|---|---|---|---|---|
| d_model | ffw_size | kv_size | n_heads | n_layers | n_vocab | Chinchilla's Reported Model Parameters (M) | Best Fit Formula's Model Parameters (M) | Standard Formula's Model Parameters (M) |
| 512 | 2048 | 64 | 8 | 8 | 32168 | 44 | 44 | 42 |
| 576 | 2304 | 64 | 9 | 9 | 32168 | 57 | 57 | 54 |
| 640 | 2560 | 64 | 10 | 10 | 32168 | 74 | 74 | 70 |
| 640 | 2560 | 64 | 10 | 13 | 32168 | 90 | 90 | 84 |
| 640 | 2560 | 64 | 10 | 16 | 32168 | 106 | 106 | 99 |
| 768 | 3072 | 64 | 12 | 12 | 32168 | 117 | 117 | 110 |
| 768 | 3072 | 64 | 12 | 15 | 32168 | 140 | 140 | 131 |
| 768 | 3072 | 64 | 12 | 18 | 32168 | 163 | 163 | 152 |
| 896 | 3584 | 64 | 14 | 14 | 32168 | 175 | 175 | 164 |
| 896 | 3584 | 64 | 14 | 16 | 32168 | 196 | 196 | 183 |
| 896 | 3584 | 64 | 14 | 18 | 32168 | 217 | 217 | 202 |
| 1024 | 4096 | 64 | 16 | 16 | 32168 | 251 | 251 | 234 |
| 1024 | 4096 | 64 | 16 | 18 | 32168 | 278 | 278 | 259 |
| 1024 | 4096 | 64 | 16 | 20 | 32168 | 306 | 306 | 285 |
| 1280 | 5120 | 128 | 10 | 18 | 32168 | 425 | 425 | 395 |
| 1280 | 5120 | 128 | 10 | 21 | 32168 | 489 | 488 | 454 |
| 1408 | 5632 | 128 | 11 | 18 | 32168 | 509 | 509 | 474 |
| 1280 | 5120 | 128 | 10 | 24 | 32168 | 552 | 552 | 513 |
| 1408 | 5632 | 128 | 11 | 21 | 32168 | 587 | 587 | 545 |
| 1536 | 6144 | 128 | 12 | 19 | 32168 | 632 | 632 | 587 |
| 1408 | 5632 | 128 | 11 | 25 | 32168 | 664 | 690 | 640 |
| 1536 | 6144 | 128 | 12 | 22 | 32168 | 724 | 724 | 672 |
| 1536 | 6144 | 128 | 12 | 23 | 32168 | 816 | 755 | 701 |
| 1792 | 7168 | 128 | 14 | 20 | 32168 | 893 | 893 | 828 |
| 1792 | 7168 | 128 | 14 | 22 | 32168 | 1018 | 976 | 905 |
| 1792 | 7168 | 128 | 14 | 26 | 32168 | 1143 | 1143 | 1060 |
| 2048 | 8192 | 128 | 16 | 20 | 32168 | 1266 | 1156 | 1073 |
| 2176 | 8704 | 128 | 17 | 22 | 32168 | 1424 | 1424 | 1320 |
| 2048 | 8192 | 128 | 16 | 25 | 32168 | 1429 | 1429 | 1324 |
| 2048 | 8192 | 128 | 16 | 28 | 32168 | 1593 | 1593 | 1475 |
| 2176 | 8704 | 128 | 17 | 25 | 32168 | 1609 | 1609 | 1490 |
| 2304 | 9216 | 128 | 18 | 24 | 32168 | 1731 | 1730 | 1603 |
| 2176 | 8704 | 128 | 17 | 28 | 32168 | 1794 | 1794 | 1661 |
| 2304 | 9216 | 128 | 18 | 26 | 32168 | 2007 | 1868 | 1730 |
| 2304 | 9216 | 128 | 18 | 32 | 32168 | 2283 | 2282 | 2113 |
| 2560 | 10240 | 128 | 20 | 26 | 32168 | 2298 | 2297 | 2127 |
| 2560 | 10240 | 128 | 20 | 30 | 32168 | 2639 | 2638 | 2442 |
| 2560 | 10240 | 128 | 20 | 34 | 32168 | 2980 | 2979 | 2756 |
| 2688 | 10752 | 128 | 22 | 36 | 32168 | 3530 | 3530 | 3257 |
| 2816 | 11264 | 128 | 22 | 36 | 32168 | 3802 | 3802 | 3516 |
| 2944 | 11776 | 128 | 22 | 36 | 32168 | 4084 | 4083 | 3785 |
| 3072 | 12288 | 128 | 24 | 36 | 32168 | 4516 | 4515 | 4176 |
| 3584 | 14336 | 128 | 28 | 40 | 32168 | 6796 | 6795 | 6281 |
| 4096 | 16384 | 128 | 32 | 42 | 32168 | 9293 | 9292 | 8587 |
| 4352 | 17408 | 128 | 32 | 47 | 32168 | 11452 | 11450 | 10613 |
| 4608 | 18432 | 128 | 36 | 44 | 32168 | 12295 | 12294 | 11360 |
| 4608 | 18432 | 128 | 32 | 47 | 32168 | 12569 | 12568 | 11680 |
| 4864 | 19456 | 128 | 36 | 47 | 32168 | 13775 | 14319 | 13266 |
| 4992 | 19968 | 128 | 32 | 49 | 32168 | 14940 | 14939 | 13937 |

| 5120 | 20480 | 128 | 40 | 47 | 32168 | 16183 | 16182 | 14950 |

Table 2: **Chinchilla Language Models.** We copy Chinchilla's Table A9 listing the model parameters and model architectural hyperparameters of all models used in the Chinchilla fitting processes. Parameters specified in millions (1e6).

## C   Theoretical Analysis

Here, we provide a detailed analysis of the empirical results obtained in the main text from a theoretical perspective. We begin by repeating the derivation of the baseline compute-optimal scaling for the number of tokens as a function of model parameters, and continue to systematically work through the perturbations discussed in section 3.

### C.1   Baseline Compute-Optimal Scaling Derivation

The Chinchilla scaling law for pretraining loss $L$ as a function of non-embedding model parameters $N$ and number of training tokens $D$ is given by eq. (4), which we repeat here

$$L(N, D) = E + \frac{A}{N^\alpha} + \frac{B}{D^\beta}, \tag{10}$$

where $E$ is the irreducible loss, and $(A, \alpha)$ and $(B, \beta)$ are parameters for the model size and data scaling terms, respectively.

The training compute budget $C$ is approximately proportional to the product of model size and training data, $C \approx cND$, where $c > 0$ is some constant factor. This allows us to express the number of training tokens as a function of compute and model size: $D = C/(cN)$. Substituting this into the loss function yields the loss for a fixed compute budget

$$L(N, C) = E + AN^{-\alpha} + B \left( \frac{C}{cN} \right)^{-\beta} = E + AN^{-\alpha} + B(c^\beta C^{-\beta})N^\beta. \tag{11}$$

The optimal model size $N_{\text{opt}}$ that minimizes the loss for a fixed compute budget $C$ is simply found by differentiating eq. (11) with respect to $N$ and setting the derivative to zero

$$\frac{\partial L}{\partial N} = -\alpha AN^{-(\alpha+1)} + \beta B(c^\beta C^{-\beta})N^{\beta-1} = 0. \tag{12}$$

Rearranging this equation reveals the optimal trade-off:

$$\alpha AN_{\text{opt}}^{-(\alpha+1)} = \beta B(c^\beta C^{-\beta})N_{\text{opt}}^{\beta-1}. \tag{13}$$

Solving for $N_{\text{opt}}$:

$$N_{\text{opt}}^{\alpha+\beta} = \frac{\alpha A}{\beta Bc^\beta}C^\beta \implies N_{\text{opt}} = \left( \frac{\alpha A}{\beta Bc^\beta} \right)^{\frac{1}{\alpha+\beta}} C^{\frac{\beta}{\alpha+\beta}} \tag{14}$$

The compute-optimal tokens-per-parameter ratio is $R_{\text{opt}} = D_{\text{opt}}/N_{\text{opt}}$. Using $D_{\text{opt}} = C/(cN_{\text{opt}})$, we find $R_{\text{opt}} = C/(cN_{\text{opt}}^2)$. Substituting our expression for $N_{\text{opt}}$ results in

$$R_{\text{opt}} = \frac{C}{c} \left[ \left( \frac{\alpha A}{\beta Bc^\beta} \right)^{\frac{1}{\alpha+\beta}} C^{\frac{\beta}{\alpha+\beta}} \right]^{-2} = \frac{C}{c} \left( \frac{\beta Bc^\beta}{\alpha A} \right)^{\frac{2}{\alpha+\beta}} C^{\frac{-2\beta}{\alpha+\beta}}. \tag{15}$$

This simplifies to the final form, which shows the ratio's dependence on the compute budget $C$ as

$$\frac{D_{\text{opt}}}{N_{\text{opt}}} = K \cdot C^{\frac{\alpha-\beta}{\alpha+\beta}}, \tag{16}$$

where $K = \frac{1}{c} \left( \frac{\beta Bc^\beta}{\alpha A} \right)^{\frac{2}{\alpha+\beta}}$ is a constant. The key insight from the Chinchilla works is that empirically, one finds that $\alpha \approx \beta$, making the exponent on $C$ approximately zero and the optimal ratio nearly constant. In mathematical terms, it leads to the relation

$$\frac{D_{\text{opt}}}{N_{\text{opt}}} \approx K = \left( \frac{B}{A} \right)^{\frac{1}{\alpha}}. \tag{17}$$

If we want to recover the $20 : 1$ ratio of training tokens to number of parameters we expect to find that $B \approx 2.85A$ for $\alpha \approx 0.35$.

## C.2    ANALYSIS OF PERTURBATIONS

We now analyze how systematic errors in model parameter counts affect the fitted scaling parameters $(\hat{A}, \hat{\alpha})$ and the resulting optimal ratio. Let $N$ be the true parameter count and $\tilde{N}$ be the perturbed (incorrect) count used for fitting. The fitting process minimizes the error between the model $L(\tilde{N}, D) = \hat{E} + \hat{A}\tilde{N}^{-\hat{\alpha}} + \hat{B}D^{-\hat{\beta}}$ and the observed losses. For most types of perturbations we consider, since the data $D$ is unaffected, we assume $\hat{B} \approx B$ and $\hat{\beta} \approx \beta$. We will break this assumption when necessary.

### C.2.1    MULTIPLICATIVE CONSTANT PERTURBATION

Assume the reported parameters are a constant multiple of the true parameters: $\tilde{N} = c_m \cdot N$. The model size term in the loss is modified to

$$\hat{A}\tilde{N}^{-\hat{\alpha}} = \hat{A}(c_m N)^{-\hat{\alpha}} = (\hat{A}c_m^{-\hat{\alpha}})N^{-\hat{\alpha}}. \tag{18}$$

For eq. (18) to match the true term $AN^{-\alpha}$, the fitting procedure will ideally find parameters such that:

- $\hat{\alpha} \approx \alpha$
- $\hat{A}c_m^{-\hat{\alpha}} \approx A \implies \hat{A} \approx Ac_m^{\alpha}$

The exponent in the optimal ratio (16) becomes $(\hat{\alpha} - \beta)/(\hat{\alpha} + \beta) \approx (\alpha - \beta)/(\alpha + \beta)$.

**Conclusion:** A multiplicative error does not change the exponent governing the trend of the optimal ratio. The flat relationship with compute budget is preserved. However, the constant prefactor $K$ is shifted by a factor of $(c_m^{\alpha})^{-2/(\alpha+\beta)} = c_m^{-2\alpha/(\alpha+\beta)} \approx c_m^{-1}$, which shifts the entire line up or down on a log-log plot. This is shown in Fig. 4 (top row) and Fig. 5 (top left).

### C.2.2    ADDITIVE CONSTANT PERTURBATION

Assume an additive error, e.g., from including/excluding embeddings: $\tilde{N} = N + c_a$. The loss term $\hat{A}(N + c_a)^{-\hat{\alpha}}$ is no longer a pure power law in $N$.

We examine the process of fitting the perturbed model $g(N; \hat{A}, \hat{\alpha}) = \hat{A}(N + c_a)^{-\hat{\alpha}}$ to the true function $f(N) = AN^{-\alpha}$ by minimizing the Mean Squared Error (MSE) in log-space.

The objective is to find $(\hat{A}, \hat{\alpha})$ that minimize $\sum_i [\log f(N_i) - \log g(N_i)]^2$. The core of the problem lies in approximating the term $\log(N + c_a)$. For the regime where $N \gg |c_a|$, which applies to the larger models in the study, we can analyze the local behavior of the function.

**Effect on the Scaling Exponent $\hat{\alpha}$:**    The most critical parameter in a power law is its exponent, which corresponds to the slope in a log-log plot. The true slope is constant

$$\frac{d(\log f(N))}{d(\log N)} = -\alpha \tag{19}$$

For the perturbed function, the effective slope is not constant and depends on $N$ as

$$\text{Effective Slope}(N) = \frac{d(\log g(N))}{d(\log N)} = -\hat{\alpha}\frac{d(\log(N + c_a))}{d(\log N)} = -\hat{\alpha}\left(\frac{N}{N + c_a}\right) \tag{20}$$

$$\implies \hat{\alpha} = \alpha / \left(\frac{N}{N + c_a}\right).$$

The fitting procedure must select a single exponent $\hat{\alpha}$ that best represents this varying slope over the range of data. To match the true average slope of $-\alpha$, the fitted $\hat{\alpha}$ must compensate for the factor $N/(N + c_a)$:

- When $c_a > 0$, the factor $N/(N + c_a) < 1$. To achieve the target slope, the fitting process must select an exponent $\hat{\alpha} > \alpha$.

- When $c_a < 0$, the factor $N/(N + c_a) > 1$ (for $N > |c_a|$). To compensate, the fitting process must select an exponent $\hat{\alpha} < \alpha$.

This provides a direct analytical explanation for the observed behavior of $\hat{\alpha}$ in Figure 4, which is smaller than the true $\alpha$ for $c_a < 0$ and increases approximately linearly for $c_a > 0$.

**Effect on the Prefactor $\hat{A}$:**  Once the optimal $\hat{\alpha}$ is determined, the prefactor $\hat{A}$ is chosen to minimize the remaining offset. We can approximate this by enforcing that the functions match at some effective "pivot" point $N_0$ that is characteristic of the dataset.

$$f(N_0) \approx g(N_0) \implies AN_0^{-\alpha} \approx \hat{A}(N_0 + c_a)^{-\hat{\alpha}}. \tag{21}$$

Solving for $\hat{A}$ gives

$$\hat{A} \approx A \cdot N_0^{-\alpha} \cdot (N_0 + c_a)^{\hat{\alpha}} = A \left( \frac{N_0 + c_a}{N_0} \right)^{\hat{\alpha}} (N_0)^{\hat{\alpha} - \alpha}. \tag{22}$$

Assuming for simplicity that the pivot point is chosen such that the $N_0^{\hat{\alpha} - \alpha}$ term is of order one, we focus on the dominant term

$$\hat{A} \approx A \left( 1 + \frac{c_a}{N_0} \right)^{\hat{\alpha}}. \tag{23}$$

This relationship explains the rapid growth of $\hat{A}$. Since we have already established that $\hat{\alpha}$ itself increases with $c$, the prefactor $\hat{A}$ grows due to two compounding effects: an increase in the base $(1 + c_a/N_0)$ and an increase in the exponent $\hat{\alpha}$. This leads to the exponential-like growth observed empirically in fig. 4.

### C.2.3 SYSTEMATIC BIAS PERTURBATION

Assume a bias where the error itself scales with model size, modeled as $\tilde{N} = \mu_{\text{geo}}(N/\mu_{\text{geo}})^s$, where $\mu_{\text{geo}}$ is the geometric mean of the true parameter counts and $s$ is the bias factor. The model size term becomes

$$\hat{A}\tilde{N}^{-\hat{\alpha}} = \hat{A} \left( \mu_{\text{geo}}^{1-s} N^s \right)^{-\hat{\alpha}} = \left( \hat{A}\mu_{\text{geo}}^{-(1-s)\hat{\alpha}} \right) N^{-s\hat{\alpha}} \tag{24}$$

To match the true term $AN^{-\alpha}$, the exponent and the constant term must satisfy the relations

$$\hat{\alpha} = \frac{\alpha}{s}, \quad \hat{A} = \mu_{\text{geo}}^{\frac{\alpha(1-s)}{s}} A. \tag{25}$$

The fitted exponent is inversely proportional to the bias factor $s$, which is verified empirically in section 3.3. The exponent in the optimal ratio is now $(\alpha/s - \beta)/(\alpha/s + \beta)$.

**Conclusion:** A systematic bias also breaks the $\hat{\alpha} \approx \beta$ condition, unless $s = 1$.

- If $s < 1$ (inflating larger models relative to smaller ones), then $\hat{\alpha} > \alpha \approx \beta$. The exponent on $C$ becomes positive, and the optimal ratio *increases* with compute.
- If $s > 1$ (shrinking larger models relative to smaller ones), then $\hat{\alpha} < \alpha \approx \beta$. The exponent on $C$ becomes negative, and the optimal ratio *decreases* with compute.

This perturbation also qualitatively changes the optimal scaling strategy, with the direction of the change depending on the nature of the bias, as seen in fig. 5 (bottom left).

## D   RELATED WORK

**Scaling Laws in Neural (Language) Models**   While initial research on scaling laws in neural models began decades ago (Barkai et al., 1993; Mhaskar, 1996; Pinkus, 1999), advances in scaling large language models brought such interest into renewed focus (Hestness et al., 2017; Kaplan et al., 2020; Brown et al., 2020), causing an explosion of research. For a non-exhaustive list, theoretical understanding of scaling laws has advanced substantially (Spigler et al., 2020; Bousquet et al., 2020; Hutter, 2021; Sharma & Kaplan, 2022; Maloney et al., 2022; Roberts et al., 2022; Bahri et al., 2024; Paquette et al., 2024; Atanasov et al., 2024; Bordelon et al., 2024a;b; Lin et al., 2024; Brill, 2024), complemented by empirical studies (Rosenfeld et al., 2020; Henighan et al., 2020; Gordon et al., 2021; Tay et al., 2021; Ghorbani et al., 2021; Zhai et al., 2022; Alabdulmohsin et al., 2022; Dehghani et al., 2023; Bachmann et al., 2023; Everett et al., 2024; Qiu et al., 2025).

Additional research has also studied how scaling interacts with specific considerations such as efficient inference (Sardana et al., 2024; Bian et al., 2025), transfer (Hernandez et al., 2021; Barnett, 2024), data quality and diversity (Chen et al., 2025; Hernandez et al., 2022; Muennighoff et al., 2023; Qin et al., 2025; Shukor et al., 2025), overtraining (Gadre et al., 2024), quantization and precision (Dettmers & Zettlemoyer, 2023; Sun et al., 2025; Kumar et al., 2024), differential privacy (McKenna et al., 2025), distillation (Busbridge et al., 2025), model architecture (Clark et al., 2022; Kudugunta et al., 2023; Abnar et al., 2025; Ludziejewski et al., 2025; Liew et al., 2025), context length (Xiong et al., 2023; Agarwal et al., 2024; Arora et al., 2024), vocabulary size (Tao et al., 2024), robustness to jailbreaking (Howe et al., 2025; Anil et al., 2024; Hughes et al., 2024), pruning (Rosenfeld et al., 2021), multimodality (Aghajanyan et al., 2023; Cherti et al., 2023), fine-tuning (Kalajdzievski, 2024; Zhang et al., 2024) and agents and world models (Pearce et al., 2025).

Recent work has also highlighted novel scaling phenomena such as inverse scaling (McKenzie et al., 2024; Gema et al., 2025), emergent capabilities (Srivastava et al., 2023; Wei et al., 2022; Schaeffer et al., 2023; Hu et al., 2024; Schaeffer et al., 2024; Snell et al., 2024b; Wu & Lo, 2024), and critical issues like data contamination (Schaeffer, 2023; Jiang et al., 2024; Dominguez-Olmedo et al., 2024) and model-data feedback loops (Dohmatob et al., 2024; Gerstgrasser et al., 2024; Kazdan et al., 2024). The advent of so-called "thinking" or "reasoning" models (Jaech et al., 2024) has sparked a new wave of interest in scaling inference compute (Brown et al., 2024; Snell et al., 2024a; Wu et al., 2024; Chen et al., 2024; Sadhukhan et al., 2025; Levi, 2025; Schaeffer et al., 2025; Kwok et al., 2025).

