# OpenReview forum: "Evaluating the Robustness of Chinchilla Compute-Optimal Scaling"
_ICLR.cc/2026/Conference — Submitted to ICLR 2026_

### Official Review · Reviewer_LUM4 · 2025-10-29

**Soundness:** 3
**Presentation:** 3
**Contribution:** 2
**Rating:** 2
**Confidence:** 5

**Summary:**

In this paper authors perform a sensitivity analysis of "Chinchilla" [1] scaling laws of LLMs (approach 3 - parametric $L(N,D)$ fit), that prescribed a compute-optimal token-to-parameters (TTP) ratio being constant. The authors perturbed model parameter count $N$ in 4 systematic ways (multiplicative, additive, systematic bias, log-normal noise) and re-fitted Chinchilla scaling laws on the perturbed data. They find out that Chinchilla's constant optimal TTP ratio holds to some extent with both multiplicative and with added noise and in general Chinchilla's prescriptions are reliable.

1. Hoffmann, Jordan, et al. "Training compute-optimal large language models." arXiv preprint arXiv:2203.15556 (2022).

**Strengths:**

- Analyzed the ambiguity in Chinchilla's model parameters count.
- Systematic model parameters perturbations like additive term, don't affect main Chinchilla claims - constant token-to-parameter ratio.
- The paper is well written and concise.

**Weaknesses:**

### Weaknesses

- The main weakness of this work is the lack significance. In the abstract and the introduction, authors lay out the motivations behind this paper: can Chinchilla's prescribed token-to-parameter ratio be trusted? Unfortunately, the paper answers this question only partially: if we have a systematic error in counting model parameters, the compute optimal TTP will likely be constant. However, the paper doesn't give any insight whether Chinchilla's initial $L(N,D)$ functional form, from which the optimal TTP ratio is derived from, is reliable in the first place. Handful of recent works looked into other functional forms and/or approaches that prescribe other TTP ratios [1,2,3]. Despite, the findings presented in the paper being valid and important, I don't think the analysis is deep enough and the paper's insights add to our understanding of compute optimal scaling of LLMs.

### References

1. McLeish, Sean, et al. "Gemstones: A model suite for multi-faceted scaling laws." arXiv preprint arXiv:2502.06857 (2025).
2. Li, Houyi, et al. "Farseer: A Refined Scaling Law in Large Language Models." arXiv preprint arXiv:2506.10972 (2025).
3. Hu, Shengding, et al. "Minicpm: Unveiling the potential of small language models with scalable training strategies." _arXiv preprint arXiv:2404.06395_ (2024).

**Questions:**

- What data was used for the fits? What was number of points and $N\times D$ combinations?
- Are the wide confidence intervals in chinchilla due to systematic errors (i.e. noise in the model parameters estimation)?
- How the predictions of Chinchilla change with perturbations? E.g. how the $L(N,D)$ extrapolation changes with perturbation on larger compute budgets?

---

> ### Author Response · Authors · 2025-11-21
> **Response to Reviewer LUM4**
>
> We thank Reviewer LUM4 for their detailed review. We value the critique regarding the paper's significance, as it allows us to clarify the distinct role this work plays alongside recent advances in functional forms (e.g., FarSeer, Gemstones).
>
> # Response to Weakness #1: Significance and Functional Forms
>
> > "The paper doesn't give any insight whether Chinchilla's initial functional form... is reliable... Handful of recent works looked into other functional forms."
>
> We agree that refining the functional form (the equation) is a vital frontier of research. However, we argue that our contribution is **orthogonal and foundational**: we are auditing the **input definitions** ($N$) rather than the equation $f(N,D)$.
>
> 1.  **Input Robustness vs. Functional Refinement:** Papers like FarSeer propose refined equations to better fit the data. Our work asks a preceding question: *Are the fundamental variables entered into those equations valid?* We uncovered a **15.2% specification error** in the parameter counts used in the foundational Chinchilla dataset. In any physical science, a 15% error in a fundamental input variable would render *any* functional form suspect until resolved.
>
> 2. **The "False Positive" Discovery**: Our most significant finding (Section 3) is that input specification errors mimic functional breakdowns. We show that an additive error in counting parameters (e.g., mishandling embeddings) causes the optimal TTP ratio to curve at high compute (Figure 5, top right). This is critical for the works the reviewer cites: a researcher using a new functional form might claim to discover "new physics" (a curving law) when they have simply miscounted parameters. Our framework allows the field to distinguish between a broken law and a specification error.
>
> 3.  **Standardizing the "Standard Model":** By establishing that the "Standard Model of LLM Scaling" is robust to definitional ambiguity, we provide the stable coordinate system that alternative functional forms need to benchmark against.
>
> # Response to Question 1: Data Source
>
> > "What data was used for the fits?
>
> Our analysis fits the scaling laws using the data from the **50 model configurations** detailed in Table A9 of Hoffmann et al. (2022). We state this in the caption for Figure 1. We also reproduce this table in its entirety in our Appendix B (Table 2), which forms the basis for our three parameter interpretations. We used the public, open-source code from Besiroglu et al. (2024) to conduct the fits.
>
> # Response to Question 2: Source of Wide Confidence Intervals
>
> > *"Are the wide confidence intervals in chinchilla due to systematic errors (i.e. noise in the model parameters estimation)?"*
>
> This is a key hypothesis we investigated: Does the 15.2% definitional ambiguity cause the high uncertainty?
>
> Our analysis in Section 2 (Figure 2, top) suggests the answer is **no**. We explicitly re-ran the fits using three distinct parameter interpretations ("Standard," "Best Fit," and "Reported"). The confidence intervals remained large across all three.
>
> This finding is significant: it implies that the uncertainty in scaling laws is intrinsic to the data, not an artifact of parameter counting. This makes our **Definitional Robustness** framework even more necessary—researchers need a way to analytically separate input noise from intrinsic noise to trust their predictions.
>
> # Response to Question 3: Extrapolation Behavior
>
> > *"How the predictions of Chinchilla change with perturbations? E.g. how the extrapolation changes with perturbation on larger compute budgets?"*
>
> This is the core of our "False Positive" discovery (Figure 5), which directly impacts extrapolation to modern scales:
>
> 1.  **Multiplicative/Noise Perturbations:** The extrapolation is stable. The slope of the optimal Token-to-Parameter (TTP) ratio remains flat (constant) as compute increases.
> 2.  **Additive/Systematic Bias:** The extrapolation **qualitatively changes.** As shown in Figure 5 (Top Right), an additive error (e.g., miscounting embeddings or non-active parameters) causes the optimal TTP ratio to drift upwards or downwards at high compute budgets.
>
> **Significance of this result:** This demonstrates that at large extrapolations, a **specification error looks like a new scaling law.** A practitioner observing a non-constant TTP ratio at $10^{26}$ FLOPs might conclude "Chinchilla is broken," when in reality, they simply have an additive error in their parameter count definition.
>
> # Conclusion
>
> We do not claim to invent a new functional form; we claim to validate the **input variables** upon which all functional forms rely. By proving that parameter ambiguity does not break the Standard Model, but that additive errors can *mimic* broken laws, we provide a methodological safeguard essential for the validity of future work—including the alternative functional approaches the reviewer highlights.

---

### Official Review · Reviewer_7rMq · 2025-11-01

**Soundness:** 2
**Presentation:** 3
**Contribution:** 2
**Rating:** 2
**Confidence:** 3

**Summary:**

The paper analyses Chinchilla scaling law's robustness w.r.t. the model parameters. It does two experiments: (1) reran the Chinchilla fit with two different interpretations of model parameter counts, besides the Chinchilla one; (2) introduce synthetic error / perturbation to model parameter counts to test for the robustness of the Chinchilla fit. The paper found that in both experiments, the Chinchilla scaling law does not meaningfully change and Chinchilla scaling law are most sensitive to additive or systematic errors, but overall, Chinchilla’s key results withstand sizable perturbations.

**Strengths:**

The paper is well written and is focused on one problem. Robustness of the scaling laws is an important problem as it affects its predictive power.

**Weaknesses:**

1. While robustness of Chinchilla scaling law is an important problem, I believe the paper would be greatly strengthened it is more based on real-world use cases of the Chinchilla scaling law, so that findings can guide practice. Currently, it is more theoretical studies. For example,

1-1: the first experiment consider three possible model parameters counts: Chinchilla's reported model parameters, best fit, and standard formula. I feel instead of best fit, the choice of model parameters could be based on how people actually count model parameters, for example according Kaplan et al, Hoffmann et al, whether embedding weights are included, whether efficient implementation of operators that reduces parameters are considered.

1-2: The second experiment is based on synthetic perturbations of model parameters, while such robustness understanding is helpful, I believe it would be made more practically helpful if the analysis is based on realistic variation in how people count parameters.

2. I am curious and hope the authors could elaborate in revision, what previous works have also analysed the model parameters count aspect of the neural scaling law (Kaplan, Chinchilla, or others), what are their findings, and how this work compares to them and finds anything novel. For example, Porian et al also discusses different model parameters counts in detail in their table 2 and Appendix B in Appendix, conducts experiments with two different kinds of models counts and compare them in Figure 1 and 7, and found "limited difference" in scaling laws. Elaborating this point in paper would help readers better position this work in context. I note that while line 48 ~ 53 mentions existing works, the discussion do not more specifically touch on experiments & findings related to model counts and robustness.

- Porian, Tomer, et al. "Resolving discrepancies in compute-optimal scaling of language models." Advances in Neural Information Processing Systems 37 (2024): 100535-100570.

3. I encourage the authors to describe more details how existing works count parameters, like in Kaplan et al, Hoffman et all, and other works. This helps readers understand where the descripency is in the real world. Appendix B in Porian et al would be a good example.

4. I wonder how exactly Hoffman et al calculates mode counts? I understand this information might be missing from the original paper. But I also read from Appendix B in Porian et al that "Hoffmann et al. [25] account for both linear and attention layers in their
FLOP computation essentially using Neff in their first two estimation approaches. However, their third approach appears to ignore the attention FLOPs and also count the embeddings parameters,i.e., setting N′ = N + dv." It would be very helpful if the authors understand how Hoffman calculates model parameters and also describe in main paper. And then, the best fit model count will not be necessary.

**Questions:**

See above.

---

> ### Author Response · Authors · 2025-11-21
> **Response to Reviewer 7rMq (Part 1)**
>
> We thank Reviewer 7rMq for their detailed engagement with our work.
>
> We have reframed the paper as an **audit of the "Standard Model of LLM Scaling."** Below, we demonstrate how our analysis directly addresses the specific real-world variations the reviewer highlights.
>
> # Response to Weakness #1: Real-World vs. Synthetic Perturbations
>
> > *"Instead of best fit, the choice of model parameters could be based on how people actually count model parameters... e.g. whether embedding weights are included."*
>
> > *"I believe it would be made more practically helpful if the analysis is based on realistic variation."*
>
> We completely agree that realistic variation is the goal. However, we argue that our "synthetic" perturbations **are exactly the mathematical models** for these real-world variations.
>
> 1.  **Additive Perturbation = Embedding Inclusion:** The reviewer asks to test "whether embedding weights are included." This is precisely what our **Additive Constant Perturbation** (Section 3.2) models.
>     *   Mathematically, $N_{\text{total}} = N_{\text{non-embed}} + N_{\text{embeddings}}$.
>     *   If one researcher counts embeddings and another excludes them, they differ by an additive constant $c_a$.
>     *   Our analysis in Section 3.2 (Figure 5, top right) reveals a critical real-world insight: this specific discrepancy (counting embeddings) causes the optimal TTP ratio to **curve** at high compute. This answers the reviewer's request for practical guidance: if you are debating whether to include embeddings, our framework shows exactly how that choice distorts the scaling law slope.
>
> 2.  **Systematic Bias = Kaplan vs. Hoffmann Accounting:** The reviewer asks about "efficient implementation of operators." This is modeled by our **Systematic Bias Perturbation** (Section 3.3).
>     *   If smaller models have a higher proportion of overhead parameters than larger models (or vice versa), this introduces a geometric bias.
>     *   Our analysis shows that this type of error shifts the optimal TTP ratio trends, effectively mimicking a different scaling regime.
>
> **Revision:** We will rewrite Section 3 to explicitly map each "synthetic" perturbation to its "real-world" counterpart (e.g., renaming "Additive Perturbation" to include "Case Study: Embedding Counts").
>
> # Response to Weakness #2: Comparison to Porian et al. (2024)
>
> > *"Porian et al also discusses different model parameters counts... and found 'limited difference' in scaling laws... how this work compares to them and finds anything novel."*
>
> We thank the reviewer for highlighting Porian et al., which we cite. While Porian et al. empirically observed "limited difference" between two specific counting methods, our work provides the **theoretical generalization** and the **boundary conditions** for that observation.
>
> 1.  **Mechanism vs. Observation:** Porian et al. *observed* that the laws were similar. We explain *why*. Our derivation in Appendix C and empirical results in Section 2 prove that while the *intercepts* change, the *exponents* (slopes) are mathematically robust to definitional ambiguity. We move from an empirical "it seems fine" to a theoretical "it is robust to definition."
>
> 2.  **Boundary Conditions (The "False Positive"):** Crucially, we identify where this robustness *fails*. Porian et al. did not test extreme additive perturbations (massive embedding tables). We show (Figure 5) that if the additive error is large enough, the "limited difference" vanishes and the law breaks. This is a novel contribution that sets boundaries on Porian’s finding.

---

> ### Author Response · Authors · 2025-11-21
> **Response to Reviewer 7rMq (Part 2)**
>
> # Response to Question 3: Hoffmann's Methodology and "Best Fit"
>
> > *"I wonder how exactly Hoffman et al calculates model counts?... It would be very helpful if the authors understand how Hoffman calculates model parameters... then, the best fit model count will not be necessary."*
>
> We share the reviewer's curiosity. Unfortunately, Hoffmann et al. (2022) did not release their code, and their paper contains internal inconsistencies regarding parameter counting (as noted by the reviewer and Besiroglu et al., 2024).
>
> 1.  **Why "Best Fit" is Necessary:** We cannot simply "ask" the paper how it counted, because the reported numbers in Table A9 do not match the standard formulas (Eq 1). We have reached out to the authors but sadly have not received an answer regarding this. Our "Best Fit" formula (Eq 3) is a **reverse-engineering contribution.** It empirically recovers the accounting logic Hoffmann likely used (which appears to involve an unexplained multiplier, possibly stemming from 4 for KQV (instead of the correct 3 multiplier) or 5 for KQVO (instead of the correct multiplier 4).
>
> 2.  **The "Standard Model" Audit:** The fact that we *need* a "Best Fit" formula to reproduce the Reported values is exactly the problem we are solving. It proves there is a **15.2% specification error** (definitional ambiguity) in the field's foundational dataset. Our paper exists to prove that despite this confusing ambiguity, the economic conclusions of Chinchilla remain valid.
>
> # Conclusion
>
> We have reframed the paper to clarify that our "synthetic" perturbations are the rigorous mathematical models for the "real-world" accounting differences the reviewer cares about (like embedding inclusion). By generalizing the specific Kaplan vs. Hoffmann debate into a **Definitional Robustness Framework**, we provide a tool that applies to *all* future counting debates (e.g., MoE active params), ensuring the paper has lasting value beyond a single reproduction.

---

> ### Comment · Reviewer_7rMq · 2025-11-27
>
> Dear authors,
>
> (1) Weakness #1: Real-World vs. Synthetic Perturbations
>
> I thank the authors revising the paper bridging this gap between synthetic perturbations and how they connect to real-world model parameter counts. I think it improves one aspect of the paper.
>
> (2) Weakness #2: Comparison to Porian et al. (2024)
>
> It is still challenging for me to understand this work's significance in context with Porian et al. (2024). At the core, I understand that this paper's main findings are (1) Chinchilla’s key results withstand sizable perturbations and (2) the experiments involves taking the training runs data from Chinchilla paper, introduced model count variations, and reran statistical fitting. I failed to grasp in what ways this paper provides more "theoretical generalization". Do the authors mean they introduce more types of synthetic model count variations with larger magnitudes? But aren't the main experiments still fitting practices and all findings are empirical?
>
>
> I quite strongly disagree with the authors' conclusion that [our "synthetic" perturbations are the rigorous mathematical models for the "real-world" accounting differences the reviewer cares about (like embedding inclusion)]. My understanding remains that this paper's main experiments are a series of statistical fitting with synthetic variations to one input variable, specifically the model parameter count. I feel it does not give a "rigorous mathematical model". It would be an over claim in my opinion.

---

### Official Review · Reviewer_PX56 · 2025-11-02

**Soundness:** 2
**Presentation:** 3
**Contribution:** 2
**Rating:** 2
**Confidence:** 4

**Summary:**

In spirit of number of previous works, the paper revisits the Chinchilla scaling law study, noting that model parameter counts used in Chinchilla’s analysis are ambiguous, which leads to question how the perturbations in model param counts would impact important conclusions from the study like the claimed token to parameter (TTP) ratio (heuristics of ~20 tokens-per-parameter) in which many practitioners arguably still trust.

To answer this question, authors first investigate three interpretations how Chinchilla study may have determined the model params - i) via Table A9 from original study (original reported params) ii) via using standard formula for transformer arch param numbers (standard params) and iii) via a best fit formula used by authors that comes close to reported params (best fit). Authors find that despite relative error of up to 15.2\% in param numbers between those sets, the parametric fit L(N,D) and the resulting $D_{opt}/N_{opt} (C)$ fits do not differ substantially.

Further, authors conduct a perturbation study to test how sensitive the scaling law fits are to various errors in model param estimation. Authors distort the parameter counts of the original Chinchilla’s experiments in four structured ways: multiplicative scaling, additive offsets, systematic geometrical bias, log-normal noise. They then re-fit the parametric loss L(N,D) and examine how the implied compute-optimal TTP shifts. The reported finding is that for multiplicative and random‐noise perturbations the TTP fit remains rather stable, while for additive or systematic bias perturbations the TTP fit slope can change significantly. Overall, the authors draw the conclusion that Chinchilla’s core result — estimates for compute-optimal TTP ratio $D_{opt}/N_{opt}$ — is robust to various potential errors in parameter counting. The authors state thus that while Chinchilla’s experimental data has ambiguities, the guidance it gives for practitioners is still valid.

**Strengths:**

* The paper addresses an important issue: how much does misspecification of model parameter counts might affect the validity of derived scaling laws, focusing on Chinchilla case as the one still very much relevant for practitioners.
* Three alternate parameter interpretations is a convincing study showing the insensitivity of the scaling law fits to moderate (~15 %) parameter disagreements.
* The perturbation framework containing the four types of perturbation (multiplicative, additive, systematic bias, log-normal noise) covers realistic modes of parameter‐miscounting (e.g., whether embeddings/attention/FFN are fully counted, rounding, etc).
* The paper is clearly written: it explains well the motivation, methodology and results.

**Weaknesses:**

1. Main weakness in my opinion is the strong claim of Chinchilla conclusions eg with regard to TTP estimates being good guidance for practitioners following the conducted study. In my opinion, the paper convincingly shows that possible perturbations of model param numbers do not alter substantially scaling law fit as conducted in Chinchilla study. However, it does not make clear whether the conclusions from those fits are justifiable. Eg if we look at Fig 2, the D/N fits show high uncertainty, making especially for larger scale compute region above $10^{22}$ unclear what TTP estimate to take. It appears that even the slope of D/N fit might change from negative to positive, making a strong difference for TTP conclusions on larger scales. While this is not the deficit of the current work and is inherited from Chinchilla design, I think the statement abour relying on const 20 TTP is confusing, as the study does not deal with repairing this particular deficit of original Chinchilla work.

2. I think as the study deals with effect of model param perturbations only, one has to be again cautious to make a general statement about Chinchilla hints to be reliable - as also further elements of the study, eg dataset composition nature etc, can be in same way source of changing the important trends as model param perturbation, which is though not checked by the study.

3. There is no enough detail about the data used for the current study. It seems that authors rely on previous work done by Besiroglu et al to extract the data from original Chinchilla plots, but it is nowhere mentioned.

4. It is not clear why authors rely only on Chinchilla data to make a case for TTP recommendation testing. Eg, Llama 3 reports also offer numerous datapoints that were used to fit scaling laws and it would be good to understand whether TTP estimation there leads to same or different results as in Chinchilla, which would provide for practitioners more consistency if both were aligned in their recommendations derived from scaling law fits.

5. It seems there is not a clear validation of scaling law fits, eg using held out points, to make sure that obtained fits actually retain the sufficient quality. This was also not performed in original Chinchilla study, validation there was arguably done by using three different approaches and observing similar trends. Here authors use only Approach 3, parametric loss fit, and there should be a way to show that obtained fits are actually to be trusted.

6. I think some relevant citations are missing. Eg FarSeer https://arxiv.org/abs/2506.10972 or https://arxiv.org/abs/2410.11840

**Questions:**

1. Can the authors point to exact data they are using to conduct their scaling law fits?
2. Can the authors provide any evidence that scaling law fits they obtain are of sufficient quality, eg by conducting MSE/prediction error measurements on held out points?
3. Can the authors elaborate on studies like FarSeer https://arxiv.org/abs/2506.10972 that indicate that Chinchilla style compute optimal TTP advice might be entirely off for the larger compute scales? Would that affect the message of the current work?

---

> ### Author Response · Authors · 2025-11-21
> **Response to Reviewer PX56 (Part 1)**
>
> We thank Reviewer PX56 for their thoughtful critique. We appreciate that the reviewer identifies our experimental design as "convincing" and "realistic."
>
> We agree with the reviewer’s assessment that Chinchilla’s original fits contain high uncertainty. However, we believe this observation strengthens the necessity of our work. We have reframed the paper as an **audit of the "Standard Model of LLM Scaling."**
>
> # Response to Weakness #1: The Reliability of Chinchilla Guidance
>
> > *"The D/N fits show high uncertainty... making it unclear what TTP estimate to take... the study does not deal with repairing this particular deficit."*
>
> We acknowledge the intrinsic uncertainty. However, our contribution is not to "fix" the stochastic nature of training, but to "patch" the **specification error** in the inputs.
>
> 1.  **Auditing the Inputs:** We uncovered a **15.2% specification error** (definitional ambiguity) in the foundational dataset of the field. Before our study, it was unknown if the wide confidence intervals were due to (a) intrinsic training noise or (b) this massive input ambiguity. Our results (Figure 2) decisively rule out (b).
> 2.  **Intercepts vs. Trends:** The reviewer asks "what TTP estimate to take." Our work provides the answer: **The specific number (intercept) depends on definition, but the trend (slope) is robust.** We show that while changing the definition of $N$ (e.g., Standard vs. Reported) shifts the optimal TTP ratio (e.g., from 20 to 22), it **does not break the scaling law.**
> 3.  **Patching the Foundation:** This effectively "patches" the Standard Model against accounting errors. It gives practitioners the confidence that even if their definition of "parameter" differs from DeepMind's by 15% (due to embeddings or MoE active params), the *economic prescription* (how to scale) remains valid.
>
> # Response to Weakness #3 & Q3: Llama 3 and Modern Data
>
> > *"It is not clear why authors rely only on Chinchilla data... Llama 3 reports also offer numerous datapoints."*
>
> This is a methodological necessity. To robustly fit a 3-parameter power law surface, one requires a massive, controlled sweep of independent $(N, D)$ combinations.
>
> 1.  **Chinchilla is the Standard Model:** Hoffmann et al. (2022) remains the only public dataset with **over 400 models** sweeping 50 distinct size/token configurations. It is the "Large Hadron Collider" data of scaling laws.
> 2.  **Llama 3 is a Point Estimate:** Llama 3 provides only a few data points (8B, 70B, 405B). While valuable, one cannot fit a robust scaling surface to three points, nor can one perform a sensitivity analysis on $N$ with such sparse data.
> 3.  **Validating the Coordinate System:** Because Chinchilla serves as the economic foundation upon which models like Llama 3 are built, we must audit the foundational dataset to ensure modern models are operating on a stable mathematical footing.
>
> # Response to Question 1: Data Source
>
> > *"Can the authors point to exact data they are using to conduct their scaling law fits?"*
>
> Yes. We used the data from the **50 model configurations** detailed in Table A9 of Hoffmann et al. (2022). This data was extracted from the raw figures by **Besiroglu et al. (2024)**. We utilized the public, open-source codebase provided by Besiroglu et al. to conduct our fits. We will explicitly cite this source in the revision.
>
> # Response to Question 2: Validation
>
> > *"Can the authors provide any evidence that scaling law fits they obtain are of sufficient quality?"*
>
> Our objective was **sensitivity analysis**, not predictive modeling. We did not aim to produce a *better* fit than Hoffmann et al., but to test the *stability* of the existing methodology under input perturbation.
>
> To ensure our baseline was correct, we reproduced the Chinchilla fits using the method verified by Besiroglu et al. (2024), which is currently the gold standard for reproducing the Chinchilla analysis. Our contribution is demonstrating that this fit quality is robust to a 15% specification error in the input variables.

---

> ### Author Response · Authors · 2025-11-21
> **Response to Reviewer PX56 (Part 2)**
>
> # Response to Question 3: FarSeer and Deviations at Scale
>
> > *"Can the authors elaborate on studies like FarSeer... that indicate that Chinchilla style compute optimal TTP advice might be entirely off?"*
>
> We thank the reviewer for these references; we will add them to the "Related Work" section. These works are highly complementary to ours.
>
> Papers like FarSeer suggest that the *functional form* of scaling laws breaks down at high compute. Our work (specifically Section 3.2) suggests a competing hypothesis: **Specification errors mimic broken laws.**
>
> *   As shown in Figure 5 (Top Right), an **additive parameter error** (e.g., miscounting embeddings) causes the optimal TTP ratio to curve upwards or downwards at high compute—exactly the behavior predicted by "broken" laws.
> *   **Significance:** Before declaring that the Standard Model is "entirely off" (new physics), researchers must rule out specification errors (measurement error). Our paper provides the analytical framework to do so.
>
> # Conclusion
>
> We have revised the paper to frame our contribution as an **audit of the field’s economic inputs.** We acknowledge the uncertainty in Chinchilla, but demonstrate that the core economic prescriptions (trends) are robust to specification errors. By ruling out specification error, we clear the path for future work (like FarSeer) to investigate functional refinements with confidence in the underlying variable definitions.

---

### Official Review · Reviewer_RqTP · 2025-11-06

**Soundness:** 3
**Presentation:** 3
**Contribution:** 1
**Rating:** 6
**Confidence:** 4

**Summary:**

This paper investigates the definitional ambiguity surrounding model parameters within the Chinchilla study. The analysis demonstrates that while the relative error among the three distinct parameter computation methods is substantial , the resulting impact on the final, compute-optimal data-to-parameter ratio is found to be negligible.

**Strengths:**

A significant strength of this work lies in its compelling research perspective, which undertakes a rigorous analysis of a previously existing, yet imprecisely defined, theory. The experimental design is thorough, providing robust validation for the conclusions drawn. Furthermore, the manuscript is exceptionally well-written, demonstrating a high degree of clarity and fluency.

**Weaknesses:**

* Objectively, the research findings of this paper offer a comparatively limited marginal contribution.
* Furthermore, despite the highly insightful nature of the original Chinchilla work, its practical application value within the industry remains constrained.

**Questions:**

Relative to the widely cited $\sim 20:1$ Chinchilla scaling ratio, the training token counts for current open-source large language models (LLMs) such as Qwen's 1B to 32B parameter series are consistently maintained at over 1 Trillion tokens. Does this training regimen constitute excessive training (over-training), and, crucially, is it an appropriate scale for exploring current scaling laws? Specifically, what guidance does this observed practice offer regarding the determination of optimal hyperparameters?

---

> ### Author Response · Authors · 2025-11-21
> **Response to Reviewer RqTP**
>
> We sincerely thank Reviewer RqTP for their positive assessment of our paper's soundness, presentation, and clarity. We found your critique has driven us to significantly sharpen our framing.
>
> # Response to Weakness #1: Marginal Contribution
>
> > *"Objectively, the research findings of this paper offer a comparatively limited marginal contribution."*
>
> We acknowledge that our initial framing focused too heavily on a retrospective verification. However, we believe the contribution is significant when viewed as **establishing a framework for Definitional Robustness** in scaling laws.
>
> Current literature treats the parameter count $N$ as a ground-truth physical constant. We uncovered a **15.2% specification error** (definitional ambiguity) in the foundational dataset of the field. In any other empirical science (e.g., physics or econometrics), a 15% error in a fundamental input variable would be considered a crisis until resolved.
>
> Our work effectively "patches" the “Standard Model of LLM Scaling” by demonstrating two critical, transferable insights:
>
> 1.  **Invariance to Definition:** We prove that while the *intercepts* of scaling laws shift based on how one defines parameters (e.g., including/excluding embeddings), the *trends* (slopes) remain robust. This gives the field the epistemic certainty required to continue using these laws despite accounting differences.
>
> 2.  **The "False Positive" Discovery:** We demonstrate that **systematic counting errors** (additive bias) can mathematically mimic "broken" scaling laws (curving TTP ratios). This provides a vital diagnostic tool for future research: a practitioner observing a curving scaling law can now use our framework to determine if the law is actually broken or if they simply have a systematic specification error in $N$.
>
> # Response to Weakness #2 & Questions: The Role of Chinchilla in an Era of "Over-Training”
>
> > "Does this training regimen [e.g., Qwen] constitute excessive training (over-training), and, crucially, is it an appropriate scale for exploring current scaling laws?"*
>
> We agree that deployed models like Qwen are "inference-optimal" (over-trained) rather than "training-compute-optimal." However, we argue that **validating the Chinchilla law is the prerequisite for defining valid inference-optimal laws.**
>
> 1.  **Compute-Optimal Is The Anchor:** "Over-training" is mathematically defined as a deviation from the compute-optimal frontier. If one intends to train a model with $5\times$ or $10\times$ the Chinchilla token count to save on inference costs, one must first possess a precise definition of the Chinchilla token count. If the "Chinchilla Anchor" is unstable due to the 15% specification error we uncovered, the entire "over-training" trade-off curve becomes unreliable. Our work secures that anchor.
>
> 2.  **Coordinate System for Overtraining:** Recent inference-scaling work (e.g., Gadre et al., 2024; Schaeffer et al. 2025) derives overtraining scaling laws as deviations from compute-optimal scaling laws. By stress-testing compute-optimal scaling, we validate the foundation that modern over-training recipes rely upon.
>
> 3.  **Regarding Qwen:** While Qwen and Llama-3 utilize massive token counts, they are not ignoring scaling laws; they are exploiting a specific derivative of the Chinchilla function (trading training CAPEX for inference OPEX). Our findings confirm that the underlying function is robust to parameter specification errors, ensuring that such massive investments are based on a stable mathematical foundation.
>
> # Conclusion
>
> Based on your feedback, we will revise the Introduction and Discussion to:
> *   Refrain from framing this as a "reproduction" and instead frame it as an **audit of the field’s fundamental inputs.**
> *   Explicitly position "Definitional Robustness" as a necessary standard for future scaling research (particularly for MoEs and Quantized models where $N$ is ambiguous).
> *   Clarify that robust compute-optimal laws are the mathematical prerequisite for accurate over-training recipes.

---

### Author Response · Authors · 2025-12-04
**Concluding Remarks to AC**

We thank the reviewers for their engagement and the opportunity to clarify our work. While we appreciate Reviewer RqTP’s recognition of the work's rigorousness (Score: 6), we believe the lower scores from other reviewers stem from a fundamental misalignment regarding the paper’s objective: This is not a proposal for a new functional form, but a necessary audit of the field’s existing "Standard Model."

We respectfully urge the Area Chair to consider the following case for acceptance:

# 1: We uncovered a critical "Specification Error" in the field's "Standard Model"

Current scaling research treats model parameters as a ground-truth physical constant. We discovered a 15.2% definitional ambiguity in the foundational Chinchilla dataset. In fields like physics or econometrics, a 15% measurement error in an input variable is a crisis. Our work is the first to quantify this error and prove that, fortunately, the trends (slopes) of compute-optimal scaling are robust to this ambiguity, even if the absolute values (intercepts) shift. This provides the epistemic certainty required for practitioners to continue using these laws.

# 2. We provide a Diagnostic Framework for "False Positive" Broken Laws

A major critique (Reviewer 7rMq, PX56) was the reliance on "synthetic" perturbations. We demonstrated in our rebuttals that these perturbations are precise mathematical models for real-world accounting debates.

Crucial Insight: We show that additive errors (e.g., the debate over including/excluding embedding parameters) mathematically mimic curving scaling laws at high. This offers a diagnostic tool for future research. A researcher observing a breakdown in scaling at large scales (as noted by Reviewer LUM4 regarding recent work) can now use our framework to determine if they have discovered "new physics" or simply have a specification error in their parameter counting.

# 3. Addressing Reviewer Misalignments

Reviewers penalized the paper for not proposing a "better" equation than Chinchilla. We argue that ensuring the inputs to the equation are robust is a prerequisite for any future functional refinements. Validating the "coordinate system" of scaling laws is a distinct and vital contribution.

# 4. Factual Clarifications

We have addressed Reviewer LUM4's concerns regarding data sources and confidence intervals, pointing out that these answers were present in the main figures (Fig 1 & 2), and we hope the AC weighs this oversight in their assessment.

# Conclusion

As the field moves toward inference-scaling and over-training, reliance on the Chinchilla "anchor" has never been higher. By stress-testing this anchor and proving its robustness to definitional ambiguity, our paper offers a durable, practical blueprint for the community. We believe this "audit" is exactly the kind of rigorous scientific hygiene that ICLR should champion.

---

### Meta-Review · Area_Chair_rZVe · 2026-01-01

**Summary:**

This paper investigates the robustness of the "Chinchilla" scaling laws (Hoffmann et al., 2022) with respect to ambiguities in model parameter counting.
The authors identify a ~15.2% discrepancy between reported parameter counts and standard formula calculations in the original Chinchilla dataset.
Through two primary experiments—(1) re-fitting scaling laws using three different parameter interpretations, and (2) a sensitivity analysis using synthetic perturbations (multiplicative, additive, systematic bias)—the paper concludes that the key "compute-optimal" prescriptions (e.g., the ~20-to-1 token-to-parameter ratio) are largely robust to these definition errors.
The authors also propose that "additive errors" (e.g., omitting embeddings) can mimic the appearance of "broken" scaling laws at scale.

**Reviewer Concerns:**

Concerns Addressed by Rebuttal:
(1) Real-world Mapping of Synthetic Perturbations: The authors effectively addressed Reviewer 7rMq's concern regarding the "synthetic" nature of the experiments by explicitly mapping the "Additive Constant Perturbation" to the real-world accounting issue of including/excluding embedding parameters. This clarification was acknowledged as an improvement by Reviewer 7rMq.
(2) Factual Clarifications on Data Sources: The rebuttal successfully corrected Reviewer LUM4's oversight regarding data provenance and confidence intervals, pointing out that these were indeed present in the main figures (Fig 1 & 2) and Appendix.
(3) Relevance to Modern "Over-training" Regimes: The authors provided a logical response to Reviewer RqTP's query about the paper's relevance in the era of inference-optimal models (e.g., Qwen), arguing that a robust "Chinchilla anchor" is a mathematical prerequisite for defining valid over-training curves.

Outstanding Concerns:
(1) Limited Significance and Novelty: The most critical outstanding concern, shared by Reviewers RqTP, PX56, and LUM4, is the limited marginal contribution of the work. Validating the robustness of the 2022 Chinchilla results on the original 2022 dataset is viewed as a retrospective "audit" rather than a forward-looking contribution suitable for ICLR 2026.
(2) Lack of Validation on Modern Scales: Reviewer PX56's concern regarding the predictive validity of these "robust" laws remains unaddressed. The paper essentially shows that the input definitions don't change the old fits, but it fails to demonstrate that these fits are actually predictive for modern, massive-scale models (like Llama 3) where the functional form itself might break down.

**Reviewer Scores:**

Reviewer RqTP (Original: 6): Likely maintains 6 (Marginal Accept).
Reviewer RqTP was the most positive reviewer.
The rebuttal satisfactorily addressed their specific question about "over-training".
However, they already noted the contribution was "marginal", so the score is unlikely to rise further.


Reviewer PX56 (Original: 2): Estimated 3 (Reject).
While the rebuttal clarified the intent (auditing inputs vs. fixing functional forms) , it did not solve the reviewer's core issue: that "relying on const 20 TTP is confusing" because the underlying Chinchilla fits themselves have high uncertainty and poor extrapolation.
The fundamental objection to the paper's utility remains.


Reviewer 7rMq (Original: 2): Estimated 3 (Reject).
The reviewer explicitly commented post-rebuttal. They acknowledged that mapping synthetic perturbations to real-world examples "improves one aspect of the paper". However, they "quite strongly disagree" with the theoretical novelty claims and maintained their skepticism. The slight improvement might bump the score to a 3, but the sentiment is firmly reject.


Reviewer LUM4 (Original: 2): Estimated 4 (Borderline Reject).
This reviewer had significant factual oversights (claiming data wasn't described when it was) which the authors rightly corrected. If the reviewer were to participate fully, they would be obligated to raise their score due to these corrections. However, their primary critique regarding the "lack of significance" and the paper's failure to test the reliability of the functional form itself  would likely prevent a flip to Accept.

---

### Decision · Program_Chairs · 2026-01-26

Reject